# OpenReview forum: "CLAMP: A Chebyshev-Weighted Multi-Gradient Approach for Multi-Objective LLM Alignment"
_ICLR.cc/2026/Conference — Submitted to ICLR 2026_

### Official Review · Reviewer_5DcQ · 2025-10-18

**Soundness:** 3
**Presentation:** 3
**Contribution:** 1
**Rating:** 2
**Confidence:** 3

**Summary:**

This paper introduces CLAMP, a multi-objective alignment framework that operates without an explicit reward model. It utilizes distinct preference datasets for various human preference dimensions to optimize a vector-valued objective function.

The framework integrates weighted Chebyshev scalarization with multi-gradient descent algorithms to find Pareto-stationary solutions. The authors also provide a theoretical guarantee for a finite-time convergence rate for the framework, which is notably independent of the number of alignment objectives. Experimental results confirm CLAMP's effectiveness in aligning LLMs to heterogeneous human preferences, showing significant improvement over existing methods.

**Strengths:**

1. The paper introduces the CLAMP framework and establishes a theoretical guarantee for solving the Multi-Objective LLM Alignment problem.

2. Experimental results validate the efficiency of the proposed method.

**Weaknesses:**

My main concern is the concept of Pareto Optimality and the Pareto Front. Though the consideration for Multi-Objective LLM Alignment across different preference functions is reasonable, the Pareto Optimality seems too weak. In detail, a Pareto optimal solution only requires that the solution is not dominated by others.

Based on this definition, if we consider a summation reward function $f(\theta)=f_1(\theta)+\dots+f_m(\theta)$ and only try to maximize this reward function, then the optimal solution for this summation reward function will naturally be Pareto Optimal. This is because, for any other solution, this optimal solution must have a larger value for at least one objective (or agent $m$) to achieve a higher summation.

In an extreme case, when one of the the function $f_i(\theta)$ is continuous, it usually does not have the same value for different solutions. In this case, only maximizing the function $f_i(\theta)$ can still find a Pareto Optimal solution.

Overall, the Pareto Optimality seems not to capture the fundamental intuition for the Multi-Objective LLM Alignment, and the calculation of the Pareto Optimal solutions can be easily reduced to the Single-Objective LLM Alignment problem, which highly challenges the contribution of this paper.

**Questions:**

1. The current experimental results are limited to LoRA-based fine-tuning. Could the authors provide results using full fine-tuning to give a more complete performance comparison?

2. The two primary multi-objective tasks ("Helpfulness-Harmlessness" and "Helpfulness-Honesty-Instruction-Following") appear similar in nature. Do the authors have any results demonstrating the divergence or degree of conflict between the objective functions in these two evaluation settings, which is essential for reflecting the difficulty of true multi-objective problems?

---

### Official Review · Reviewer_y3m2 · 2025-10-28

**Soundness:** 3
**Presentation:** 3
**Contribution:** 3
**Rating:** 4
**Confidence:** 3

**Summary:**

This paper proposes CLAMP (Chebyshev-Weighted Multi-Gradient Alignment), a new framework for multi-objective LLM alignment that does not rely on reinforcement learning or reward models. The method addresses the challenge of aligning models with multiple, potentially conflicting objectives by formulating training as a multi-objective optimization problem.

CLAMP defines a Chebyshev-weighted loss, which minimizes the maximum deviation among all objectives, effectively prioritizing the worst-performing one at each step. The optimization combines this scalarization with the Multi-Gradient Descent Algorithm (MGDA) to compute a single update direction that achieves Pareto-stationary improvements across objectives. The approach includes theoretical analysis proving an (O(1/T)) convergence rate independent of the number of objectives, indicating good scalability.

Empirically, CLAMP is tested on multi-preference alignment benchmarks and compared with MORLHF and related baselines. Results show improved Pareto front coverage, alignment stability, and task trade-offs with minimal computational overhead. The paper claims CLAMP offers a theoretically principled and efficient alternative for balancing multiple alignment goals in LLMs.

**Strengths:**

The paper addresses a problem in multi-objective alignment for LLMs and proposes a RL-free framework that is both theoretically motivated and empirically supported. In terms of originality, the integration of Chebyshev-weighted scalarization with multi-gradient descent (MGDA) offers an interesting combination of classical multi-objective optimization principles and modern LLM alignment methods. The formulation provides a clear geometric interpretation of balancing conflicting objectives and offers an alternative to traditional RLHF-based approaches.

Regarding quality, the paper includes formal convergence analysis and claims an O(1/T) convergence rate independent of the number of objectives, suggesting theoretical soundness and scalability. The optimization strategy is simple yet mathematically grounded, and the experimental evaluation demonstrates that CLAMP can achieve balanced alignment across objectives while maintaining low computational overhead.

In terms of significance, the method contributes to a growing line of research aiming to reduce reliance on RL and reward modeling in preference alignment. The focus on Pareto-stationary updates aligns well with real-world alignment challenges where multiple preferences must coexist. Although the implementation could be clarified, the framework itself has potential for broader applicability in multi-objective fine-tuning of LLMs.

**Weaknesses:**

1. While the paper introduces a theoretically motivated framework, several issues limit its clarity and empirical strength. First, the core loss formulation in Equation (3) appears mathematically equivalent to the MaxMin-RLHF objective when incorporating the preference vector p. However, the paper only compares with MORLHF and does not include MaxMin-RLHF. This omission makes it difficult to assess whether CLAMP’s improvements arise from the algorithm itself or simply from reparameterization of an existing loss.

2. The proposed loss function does not have a unique solution. The Chebyshev-weighted max-min scalarization naturally admits multiple Pareto-stationary points, depending on initialization and gradient geometry. The paper does not provide any analysis or experiments on how these different solutions behave in practice. Multiple training runs could yield models representing distinct trade-offs on the Pareto front, but the paper lacks results or visualizations demonstrating this diversity or consistency. Clarifying how these solutions differ and whether they produce meaningful alignment trade-offs would substantially strengthen the empirical section.

**Questions:**

1.  Comparison with MaxMin-RLHF and Computational Advantage: Equation (3) appears to define the same loss function as the MaxMin-RLHF formulation when incorporating the preference vector $p$, i.e., minimizing
$\min_\theta \max_m \{ p_m f_m(\theta) \}$.
However, the paper only compares CLAMP with MORLHF rather than with this closely related MaxMin-RLHF method, which shares the same scalarized objective. Could the authors clarify how CLAMP provides a meaningful improvement, either theoretically or empirically, over MaxMin-RLHF, given that both methods optimize an equivalent objective but differ in optimization dynamics?

2. Could the authors include any experimental results or analysis comparing multiple runs to demonstrate whether the algorithm consistently converges to similar solutions or explores diverse regions of the Pareto front? The loss function defined in Equation (3) does not appear to yield a unique optimal solution, as multiple Pareto-stationary points can exist depending on initialization or gradient geometry. It would also be helpful to visualize or quantify the diversity of solutions obtained from different random seeds to confirm the robustness and stability of CLAMP’s optimization process.

---

### Official Review · Reviewer_GDJ6 · 2025-10-30

**Soundness:** 2
**Presentation:** 2
**Contribution:** 2
**Rating:** 4
**Confidence:** 3

**Summary:**

This paper proposes CLAMP (Chebyshev-weighted LLM alignment with multi-objective preferences), a method that integrates stochastic multi-gradient-based and Chebyshev-weighted techniques to achieve multi-objective alignment for LLMs. Experimental results demonstrate that the proposed approach improves multi-objective alignment compared to existing baselines.

**Strengths:**

- The proposed method is theoretically grounded.

- The method is RL-free and reward model-free, and its training time remains unaffected by the number of objectives.

**Weaknesses:**

- The clarity of the paper needs improvement. It took me some time to understand the methodology, and after reading, it remains unclear how to compute the multi-objective loss. For example, given a single sample (x, y_w, y_l) to optimize and three objectives, how is the loss function computed for each objective?

- The novelty appears limited, as the method primarily applies previous theories to the multi-objective alignment problem.

- Comparing with more recent baselines, such as those mentioned (e.g., MO-GRPO) in the related work, would enhance the overall quality of the paper.

- The term "heuristics" could be misleading. Are all existing multi-objective alignment algorithms heuristic-based? The authors should provide a clearer explanation of this.

- The proposed method is sensitive to the hyperparameter μ.

- The experimental settings are not clearly stated. For instance, the test set and the reward models used should be explicitly specified, with direct references to the appendix where applicable.

- The paper does not validate whether the proposed algorithm performs well with other methods, such as IPO or SimPO.

- A concern is the notably poor performance of MORLHF reported in the paper. In my own experience, using multi-objective reward-weighted PPO/GRPO often yields better results than DPO. Could the weak performance be due to the reward models used? It would be beneficial if the authors could report the results of training with more advanced reward models, such as Skywork-llama3-8b-v2 on UltraFeedback.

- Formatting issues: The citation format is incorrect. For example, "Reinforcement learning from human feedback (RLHF) Christiano et al. (2017);

- Typo:  Line 232: “meta-algorithm.” should be “meta-algorithm”.

**Questions:**

- Can CLAMP be applied to the online RL setting?

---

### Meta-Review · Area_Chair_b5Q2 · 2026-01-07

**Summary:**

This paper proposes CLAMP (Chebyshev-weighted LLM alignment with multi-objective preferences), a new RL-free and reward-model-free algorithmic framework. However, the theory used in the paper is challenged by reviewers and the novelty is limited. Since no rebuttal is provided by the authors, AC recommends rejection.

**Reviewer Concerns:**

The author did not submit any discussion or rebuttal. So all problems are still unresolved.

**Reviewer Scores:**

The author did not submit any discussion or rebuttal, and it is not likely that any reviewer would have changed their scores.

---

### Decision · Program_Chairs · 2026-01-26

Reject